# Associations between Infant Dietary Intakes and Liking for Sweetness and Fattiness Sensations in 8-to-12-Year-Old Children

**DOI:** 10.3390/nu13082659

**Published:** 2021-07-30

**Authors:** Wen Lun Yuan, Sophie Nicklaus, Anne Forhan, Claire Chabanet, Barbara Heude, Marie-Aline Charles, Christine Lange, Blandine de Lauzon-Guillain

**Affiliations:** 1Center des Sciences du Goût et de l’Alimentation, AgroSup Dijon, CNRS, INRAE, Université Bourgogne Franche-Comté, F-21000 Dijon, France; sophie.nicklaus@inrae.fr (S.N.); claire.chabanet@inrae.fr (C.C.); christine.lange@inrae.fr (C.L.); 2Université de Paris, CRESS, Inserm, INRAE, F-75004 Paris, France; anne.forhan@inserm.fr (A.F.); barbara.heude@inserm.fr (B.H.); marie-aline.charles@inserm.fr (M.-A.C.); blandine.delauzon@inserm.fr (B.d.L.-G.)

**Keywords:** food liking, children, dietary intakes, infancy, longitudinal study, EDEN cohort

## Abstract

An exposure to sweetened and fatty foods early in life may be involved in high liking later in life. The objective is to investigate the association between dietary exposure to carbohydrate, sugars and fat in infancy, with liking for sweetness, fattiness and fattiness-and-sweetness sensations at 8-to-12-year-old. Analyses were conducted on 759 French children from the EDEN mother-child cohort. Carbohydrate, sugar or fat intake, being a consumer of added sugars or added fats were assessed at 8 and 12 months using 3-day food records. The liking score (0–10) for the different sensations was assessed through an online child-completed questionnaire. Associations were tested by linear regressions adjusted for main confounders and the interaction with sex was tested. None of the early dietary exposure variables was related to fattiness liking. Carbohydrate intake at 8 months was positively but weakly associated with liking for sweetness-and-fattiness. In girls only, carbohydrate intake at 12 months was positively associated with liking for sweetness. Globally, no marked associations were observed between infant dietary exposure to sweet and fat and liking for sweetness and fattiness in young children. The positive link in girls between early carbohydrate exposure and later liking for sweetness needs to be confirmed in further studies.

## 1. Introduction

According to the “developmental origin of health and disease” hypothesis, an unbalanced diet in early life may contribute to adverse cardiometabolic outcomes later in life [1]. A body of evidence has shown that diet quality in infancy could influence food preferences [2,3,4] and food intake [5,6,7] later in childhood because of both the tracking of dietary patterns throughout childhood and early learning mechanisms [8]. Unhealthy dietary patterns, characterized by a frequent consumption of processed fatty and sweetened foods, have already been observed as soon as infancy in various studies (for instance, “Western-like” [9], “adult foods” [10], “biscuits, sweets and crisps” [11], “processed and fast foods” [5], and “noncore foods” patterns [12]). In this context, it is important to understand the degree at which the hedonic dimension related to food experience (e.g., food preference, food liking) is influenced by early dietary exposure [13].

Taste preference is usually referring to the comparison of individual reaction to different concentrations of tastants (sucrose, oleic fatty acid.) or to a tastant relative to water [14,15,16,17,18,19], whereas taste liking is rather referring to the degree of appreciation towards a food taste on a hedonic scale [20,21,22,23,24,25]. As an example, two individuals can both prefer high level of sweetness to low level of sweetness; but one can rate degree of liking for high level of sweetness higher than the other [26].

In contrast to liking for sweetness which is innate [27], liking for fattiness is not congenital but rather learned through the association between the ingestion of fatty foods and its positive physiological consequences (satiety) [28,29,30,31,32,33]. However, whether high levels of sugar intake in infancy could contribute to a high liking for sweetness in later childhood is ambiguous [34]. Previous findings suggested that repeated exposures to sweetened foods can be part of the learning process of liking for sweetness [16,35,36]. A longitudinal study conducted in the US, has shown that repeated exposures to sweetened beverages during infancy could lead to a preference for solutions with higher concentrations of sucrose later in childhood [16]. In a randomized trial, exposure to a slightly sweet supplement from 6 to 18 months was neither associated with later preference, consumption of sweet foods and beverages [15] nor with the level of sweet taste most preferred in 4- to 6-year-old Ghanaian children [14]. The authors hypothesized that the sugar content in their supplement could have been too low to shift the children’s behavior towards sweetened foods at later stages [14,15] compared with the previous study on frequent consumption of sweetened beverages [16], because of the highly and routine degree of exposure to sweetened foods, in particular sweetened beverages during the first two years of life [14,15,37]. However, this trial did not investigate the influence of overall dietary exposure to sugar during infancy on later liking for sweetness in young children but only the influence through the supplementation. Similarly, the American study only investigated the influence of consumption of sweetened beverages in infancy, but not the total sugar intake [16]. Hence, there is no evidence on whether infant overall sugar intake could influence later liking for sweetness. Furthermore, except for the two aforementioned longitudinal studies, there is a lack of evidence on the long-term association between early dietary exposure to sugar and young children sweetness preference. Therefore, there is a need to investigate further this association in prospective studies and in population settings with different level of sugar exposure. Concerning fattiness, this sensation is contributing to food palatability which can contribute to overconsumption [38], but the influence of early exposure on liking for fattiness sensation was not documented.

Sex-related differences in dietary behavior have already been noted in children, regarding both food intake [39,40] and food preferences [39,41,42,43]. Previous evidence suggested that infant feeding practices differed by sex. In France for instance, girls were more likely to be introduced later to complementary food, breastfed longer and consumed more ready-prepared baby foods than boys [44,45,46], which may lead to a lower exposure to sweetness and fattiness in infancy [40]. Furthermore, girls tended to consume more and have greater liking for fruit and vegetables than boys [39,41,42,43]. In contrast, boys had higher liking for sweetened and fatty foods than girls [41]. Neural responses related to food cues were also evidenced in children. Boys showed greater activation to visual food cues in the right hippocampus and visual cortex than girls [47]. In contrary, girls might be more sensitive to sweet taste [48,49]. In other words, the same dietary exposure among boys and girls could be perceived differently and lead to different behavioral reactions depending on sex.

The present study aimed to investigate the associations between infant dietary exposure to carbohydrate, sugars and fat and liking for sweetness, fattiness and fattiness-and-sweetness sensations in 8-to-12-year-old children, accounting for a potential moderating effect of sex on the associations. We hypothesized that a high intake of carbohydrate and fat may lead to a high liking for sweetness, fattiness and fattiness-and-sweetness sensations and that these associations may differ by sex.

## 2. Materials and Methods

### 2.1. Study Population

The present study was based on the EDEN (“Etude des Déterminants pre- et post-natals de la santé de l’ENfant”) mother-child cohort study. A detailed study protocol was published previously [50]. Briefly, the EDEN study included 2002 pregnant women during their hospital visit before 24 weeks of amenorrhea, between 2003 and 2006, in two cities in France (Nancy and Poitiers). The exclusion criteria included multiple pregnancy, known diabetes before pregnancy, French illiteracy and planning to move outside the region within the next three years. Written consent was obtained from the parents.

For the present study, data about consumption in infancy (when children were aged 8 or 12 months) were used. Moreover, families accepting to pursue participation beyond the initially planned follow-up were contacted to let their children complete an online questionnaire on food liking. Between May and September 2015 (when the children were aged 8 to 12 years), parents were invited to participate either by electronic mail (78%) or by paper mail (22%), when no valid electronic mail address was available.

### 2.2. Infant’s Dietary Intake

Infant food intake was reported on a daily basis (according to the instructions) by mothers using a 3-day food record including two weekdays and one weekend day when their child was aged 8 and 12 months. The food record included information about the provided and the uneaten amount of food, the food brand name or the recipe for home-made foods. Nutrient intake was calculated based on food composition databases from the French baby foods industry group (SFAE 2005) and from the French Observatory of Food Nutritional Quality (CIQUAL 2020). Carbohydrate, sugar, fat and energy intake were calculated only among non-breastfed infants, as breast milk intake could not be measured.

No information about free sugars was available in the food composition databases. However, consumption of natural sweeteners (honey, white sugar, brown sugar, jam), sweetened beverages (sodas, fruit juices) were collected from food records. Sweetened beverages and natural sweeteners were grouped into added sugars group. The consumption of added fats (vegetable oils, animal fats) was also collected. Infants were categorized as consumers or non-consumers of added sugars or added fats.

### 2.3. Children’s Liking for Sweetness and Fattiness Sensations

Children’s liking for sweetness and fattiness sensations was assessed when they were aged between 8 and 12 years by a specific and validated questionnaire. The liking questionnaire, completed by children, was initially developed for adults [51] but adapted and internally validated for children [24]. This questionnaire assessed liking for saltiness, sweetness and fattiness sensations which is itself composed of liking for fattiness-and-saltiness and fattiness-and-sweetness sensations. In the current study, liking for saltiness was not investigated.

Liking for sweetness and fattiness was built on 54 items divided into 3 sections. Section 1 assessed liking for sweet (i.e., fruit juices, honey), fatty-salty (i.e., chips, sausages) and fatty-sweet foods (i.e., doughnut, chocolate croissant). Liking was rated on a 9-point hedonic scale from “I really don’t like at all” to “I really like very much”. Subjects could also choose “I have never tasted this food”. Section 2 evaluated preferences for different levels of sweet, fat-and-salt or fat-and-sweet seasonings. Preferences for seasonings were assessed on a 5- or 6-point scale from “without” to “with a lot of”. Finally, Section 3 questioned about dietary behavior in terms of sweet and fatty foods. Questions regarding dietary behavior were measured on a 5-point frequency scale from “never” to “always” or a 9-point scale from “not at all” to “a lot”. For Section 2 and Section 3, subjects could also answer “I do not like this food”. Answers “I have never tasted” (Section 1) or “I do not like this food” (Section 2 and Section 3) were considered as missing data.

Liking for sweetness, fattiness and fattiness-and-sweetness, had a singular factor structure composed of multiple latent constructs (factors), themselves composed of several items. Liking for sweetness (16 items) was composed of 3 latent constructs: liking for sweet foods, liking for added sugars and liking for added jam. Liking for fattiness was composed of liking for fattiness-and-saltiness sensation and by liking for fattiness-and-sweetness sensation. Liking for fattiness-and-saltiness sensation was built on 3 latent constructs: liking for added fats and salt (9 items), liking for cheesy foods (3 items) and liking for fatty-salty foods (10 items). Liking for fattiness-and-sweetness sensation was based on 3 latent constructs: liking for added whipped cream (5 items), liking for fatty-sweet foods (9 items) and liking for added chocolate spread (2 items). Using the previously published factor structure [24], goodness-of-fit indices of all factor structures were tested and considered acceptable in the study sample (liking for sweetness: root mean square error of approximation (RMSEA) = 0.05, confirmatory fit index (CFI) = 0.92, non-normed fit index (NNFI) = 0.90; liking for fattiness sensation: RMSEA = 0.04, CFI = 0.85, NNFI = 0.84; liking for fattiness-and-sweetness sensation: RMSEA = 0.05, CFI = 0.94, NNFI = 0.92). A liking score, ranging from 0 to 10 points, was calculated for each sensation as the mean of the latent construct scores, themselves calculated as the mean of the (non-missing) contributing items. Liking scores were calculated only in children who answered that they knew and consumed at least 75% of the food items (representing at least 8 non-missing items for liking for sweetness or fattiness-and-sweetness sensation and at least 29 non-missing items for liking for fattiness).

### 2.4. Familial and Children Characteristics

Maternal characteristics (age at delivery, pre-pregnancy BMI, educational attainment, household income, parity) were collected during a face-to-face interview between 24 and 28 weeks of amenorrhea. Information about the children sex, birth weight, and gestational age was collected from the obstetrical and pediatric records. Children weight and height information were reported at the 8–12-year follow-up as a sub-part of the liking questionnaire. Child’s overweight (including obesity) was defined according to the IOTF cut-offs [52].

Maternal diet during the last trimester of pregnancy was assessed using a validated food frequency questionnaire (FFQ), completed at birth [53]. Maternal dietary patterns during pregnancy were previously identified by principal component analysis (PCA) [54]. Two main patterns were identified: (a) a “Healthy pattern” (characterized by a high intake in fruit, vegetables, fish, and whole grain cereals) and (b) a “Western pattern” (characterized by a high intake in processed and snacking foods).

### 2.5. Infant’s Feeding Practices

Any breastfeeding duration and age at complementary food introduction were calculated from the data collected at ages 4, 8 and 12 months [44]. Breastfeeding duration was categorized as “never breastfed”, “less than 3 months”, “3 to 6 months” and “at least 6 months”. Age at complementary food introduction were calculated from the data collected at ages 4, 8 and 12 months [44]. Age at complementary food introduction was categorized as “before 4 months”, “between 4 to 6 months”, “after 6 months”.

### 2.6. Sample Selection

The different steps of the sample selection process are described in Figure 1. Among the EDEN participants, 785 children filled out the liking questionnaire. Only the children who answered that they knew and consumed at least 75% of the food items were considered in the present study, leaving 759 children. Among the 759 children, only those with nutrient intake assessed at 8 or at 12 months (excluding children with missing or invalid food records (*n* = 60 and *n* = 113, respectively) and children no more breastfed (*n* = 83 and *n* = 36, respectively)) were kept in the main analysis (*n* = 616 and *n* = 610, respectively). Complete-case analysis at 8 and 12 months was run on 555 and 543 children respectively, excluding children with missing data for covariates and confounders at 8 and 12 months (*n* = 61 and *n* = 67).

### 2.7. Statistical Analysis

Student’s *t*-test and Chi-squared test were used to compare the characteristics of included and non-included participants; and the dietary intakes and the liking scores of boys and girls. Paired Student’s *t*-test was used to compare the liking scores between sweetness, fatty and fattiness-and-sweetness sensations.

Unadjusted and adjusted associations between dietary exposure variables in infancy and liking outcome variables in young children were tested either using linear or logistic regressions. Each dietary intake variable (carbohydrate, sugar or fat intake, being a consumer of added sugars or added fats) was tested individually, i.e., not concomitantly with another dietary variable within the same model. Carbohydrate intake, sugar intake and consumption of added sugars were consecutively analyzed in association with liking for sweetness sensation. Fat intake and consumption of added fats were successively investigated in relation with liking for fattiness sensation. Finally, carbohydrate, sugar or fat intake, being a consumer of added sugars or added fats were consecutively studied in association with liking for fattiness-and-sweetness sensation. For all the studied associations, potential confounding factors were identified in the literature and selected using the directed acyclic graph (DAG) method [39]. Hence, the following set of confounding variables were considered for all models: study center, children’s characteristics (sex, age, birthweight, gestational age), maternal characteristics (age, pre-pregnancy BMI, educational attainment, household income, parity and maternal scores on healthy and western dietary patterns during pregnancy), any breastfeeding duration and age at complementary food introduction.

Missing values for confounders, which were assumed to be missing at random, were handled using multiple imputations. Five independent datasets were generated using the Markov chain Monte Carlo method (PROC MI with FCS LOGISTIC and FCS REGPMM statements) and pooled effect estimates were calculated (PROC MIANALYZE). The multiple imputation method is detailed in the Appendix A. Complete case analysis was performed as a sensitivity analysis to assess the robustness of our associations. The interactions between child’s sex and dietary exposure were tested in the adjusted models and imputed dataset (considered as our main analysis). When an interaction was significant, the association was examined separately among boys and girls in the unadjusted model, as well as in the adjusted models based on both complete-case and imputed samples.

All analyses were performed with SAS software (version 9.4; SAS Institute, Cary, NC, USA). Significance was set at *p* < 0.05 except for interaction tests (*p* < 0.10).

## 3. Results

Sample characteristics are presented in Table 1. There was no substantial difference between the characteristics of the sample at 8 months and at 12 months. Hence, only characteristics in children with liking scores at 8–12-year-old and nutrient intakes assessed at 12 months will be further described. In the present study, children were aged from 8.7 to 12.5 years, with a mean age of 10.7 years. Among children with a recorded weight and height in the liking questionnaire, 10% were overweight (including obesity) according to IOTF cut-offs [55]. Among EDEN participants, excluded children with no liking questionnaire or no nutrient assessed at 12 months (*n* = 1297), compared with included children (*n* = 610), were more likely to be born to a mother who is younger (29 ± 5 vs. 30 ± 5 years, *p* < 0.0001), with higher rate of obesity (10% vs. 6%, *p* = 0.01), an educational level lower than a high school diploma (35% vs. 14%, *p* < 0.0001), a household income lower than €1 500/month (22% vs. 6%, *p* < 0.0001). Excluded children were also more likely to be never breastfed (29% vs. 23%, *p* = 0.007) and introduced to complementary foods before 4 months (33% vs. 25%, *p* < 0.0001).

The liking for sweetness, fattiness and fattiness-and-sweetness sensation scores were 4.6 ± 1.3, 5.7 ± 1.2 and 5.8 ± 1.7, respectively (Table 2). The liking for sweetness score was lower than the liking for fattiness sensation score (*p* < 0.0001) and the liking for fattiness-and-sweetness sensation score (*p* < 0.0001). Compared with girls, boys had higher carbohydrate intake at age 8 and 12 months and higher liking for sweetness in young children. The percentage of added sugar consumers at age 8 and 12 months did not differ between sex even though boys had higher sugar intake at 12 months (but not at 8 months) than girls. Boys also had higher fat intake at age 12 months (but not at 8 months) and higher liking for fattiness and fattiness-and-sweetness sensations at 8 months. No difference in the percentage of added fats consumer was observed between sexes.

### 3.1. Liking for Sweetness

Few associations were highlighted between dietary intake in infancy and liking for sweetness sensation in young children. Being a consumer of added sugars at ages 8 and 12 months was not related to the liking for sweetness score and no interaction with sex was observed at 8 months and 12 months (*p*_interaction_ = 0.14 and 0.33, respectively) (Table 3). Regarding carbohydrate and sugar intake at 8 months, neither association with liking for sweetness nor interaction with sex were observed (*p*_interaction_ = 0.66 and 0.84, respectively). The association between carbohydrate or sugar intake at 12 months and the liking for sweetness score differed by sex (*p*_interaction_ = 0.003 for carbohydrate intake and *p*_interaction_ = 0.07 for sugar intake). A significant, positive but weak association was found between carbohydrate intake and liking for sweetness in 8-to-12-year-old girls; and a negative trend was observed in boys. For sugar intake, only a positive trend was observed in girls. Directions of the associations remained unchanged in the complete-case analysis.

### 3.2. Liking for Fattiness Sensation

Neither being a consumer of added fats nor fat intake at ages 8 or 12 months were associated with the liking for fattiness sensation score in young children (Table 4). No interaction with children’s sex was observed (all *p*_interaction_ > 0.50). Similar findings were found in the complete-case analysis.

### 3.3. Liking for Fattiness-and-Sweetness

Overall, the dietary intake variables at ages 8 and 12 months were not related to the liking for fattiness-and-sweetness sensation score in young children, except for a positive link between carbohydrate intake at age 8 months and the liking for fattiness-and-sweetness sensation score (Table 5). No interaction was found with children’s sex (for fat intake, at 8 months *p*_interaction_ = 0.26 and all other *p*_interaction_ > 0.60). The complete-case analysis showed similar trends.

## 4. Discussion

In the present study, no marked associations between infant dietary exposures to sugars and fat and liking for sweetness, sweetness-and-fattiness or fattiness sensations in 8-to-12-year-old children were observed. Positive links were observed, between carbohydrate intake at age 12 months and later liking for sweetness in girls only; and between carbohydrate intake at age 8 months and later liking for fattiness-and-sweetness sensation.

Previous studies found inconsistent associations between exposure to sweetness and liking for sweetness sensation. In fact, American children frequently exposed to sweetened water during infancy were more likely to prefer solutions with higher concentrations of sucrose at ages 6–10 years compared with children rarely exposed to sweetened water in infancy [16]. In an 8-day trial, a repeated daily exposure to a sweet drink with a high concentration of sucrose in 9-year-old children was associated with an increase in their preference for this sweet drink [19]. However, other studies did not find such associations [14,15,20]. The level and frequency of exposure may contribute to the observed disparities between studies. In fact, in a randomized trial in Ghanaian preschool children, a daily consumption of a slightly sweet lipid-based supplement from 6 to 18 months was neither associated with an increased preference for or consumption of sweet foods and beverages [15], nor with an increase in the level of sweet taste most preferred [14]; but the authors did not rule out that the level of exposure in their controlled trial was too low to observe a shift in later liking for sweetness between supplemented and non-supplemented children [14,15]. In French children aged 7–12 years, a contemporary frequent consumption of sweet drinks or added sugars was not associated with liking for sweetness sensation [20], but both sweet drinks and added sugars were consumed less than 2 to 3 times per week [20]. Similarly, in the present study, the consumption of added sugars (including sweetened beverages) in infancy was not associated with later liking for sweetness (or fattiness-and-sweetness) sensations in young children, but we were not able examine the frequency of added sugar consumption. Using carbohydrate intake, a weak positive association was observed in girls between carbohydrate intake at age 12 months and liking for sweetness in young children. A similar trend was observed for sugar intake in girls. Additionally, carbohydrate intake at 8 months could be positively linked to a liking for fattiness-and-sweetness sensation as well. The lack of association in our study could be partly due to the relatively low and homogeneous level of exposure in infancy in our study sample (about 60 and 70 g per day of sugar intake at 8 and 12 months, respectively and about 30 g per day of fat intake at both 8 and 12 months). Further studies have to be conducted in population with larger variation in exposure level and probably higher consumption of added sugars and fats.

Sex differences in children’s diet, food preferences and feeding practices were previously observed [39,40,41,42,43,44,45,56]. Interestingly, in the present study, boys had higher liking for sugary foods (and for fatty foods) and higher carbohydrate intake at 12 months compared with girls. This is consistent with reported higher carbohydrate intake in boys compared to girls among older children in the US [57]. As a potential explanation of our findings, it was previously suggested that girls might be more sensitive to sweet taste [48]. Hence, given their heightened sensitivity to sweetness, girls exposed to a relatively lower sweet level during infancy, could be more prone to increase their later liking for sweetness than boys.

Regarding liking for fattiness sensation or fattiness-and-sweetness sensation, we did not find any association with fat intake during infancy. Consistently, in a cross-sectional study, a frequent consumption of fatty foods was not related to the preference for foods with higher fat in children aged 6 to 9 years [18]. In contrast, at the same ages, short-term repeated exposure to a diet with a high fat content was associated with the preference for fattier foods in children [25]. Similarly, in adolescents, fat intake was associated positively with liking for high fat and high sugar milkshake over 4 years [23]. Previous associations did not distinguish liking for overall fattiness and fattiness-and-sweetness sensations. Furthermore, no study examined prospectively the influence of early dietary intakes on liking for fattiness and fattiness-and-sweetness sensations in young children, which limits comparisons with our findings.

In the present study, infant dietary intake was assessed by 3-day food records which were checked a posteriori and validated by a dietician. As we were not able to directly isolate free sugar intake from total simple sugar intake, we identified consumers of added sugars, but only one third of the sample consumed added sugars. Later dietary exposure to sweetened foods, added sugars and/or sugars in childhood could have a more impactful contribution to children liking for sweetness sensation than exposure during the first year. Regarding children’s liking, it was assessed by children themselves, which limits reporting bias compared with an indirect measurement through parental report. This measurement is not comparable with an actual tasting test since it is based on their representation of the food items and on their eating habits [24]. Nevertheless, using the adult version of this questionnaire, the risk of being overweight among adults was associated negatively with sweetness sensation liking and positively with fattiness and fattiness-and-sweetness sensation liking [21,22].

As in most cohort studies, mothers from our study sample were relatively more frequently highly educated and had higher household incomes than the national average. This could explain the low level and the lack of variability in the infant dietary exposure to sugar and fat as such feeding practices vary according to a socioeconomic gradient [44,45]. Consequently, caution needs to be taken about the generalization of our results. However, this is the first study using a longitudinal design with a long follow-up and a large sample size that investigated the association between a dietary exposure to sweetness and fattiness in infancy and liking for sweetness and fattiness sensations in young children. Elucidating whether a liking for sweetness and fattiness sensations may track into adulthood and assessing the relationship between liking for these sensations and later health outcomes related to cardiometabolic risk should be of great interest.

## 5. Conclusions

To conclude, this study did not show marked associations between dietary exposure to fat and sugar in infancy and liking for sweetness and fattiness sensations in 8-to-12-year-old children. In girls, the exposure to carbohydrate at age 12 months was positively related to the liking for sweetness; and the exposure to carbohydrate at age 8 months was positively related to the liking for fattiness-and-sweetness. More studies, ideally conducted in populations with a wider variability and a more specific characterization of the exposure to sugar and fat in infancy are needed to further explore these associations.

## Figures and Tables

**Figure 1 nutrients-13-02659-f001:**
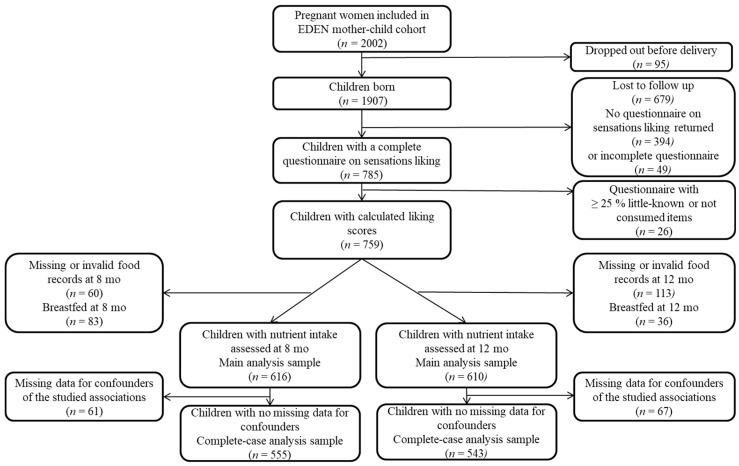
Sample selection flow chart.

**Table 1 nutrients-13-02659-t001:** Sample characteristics ^1^.

	8 Months (*n* = 616)		12 Months (*n* = 610)	
	*n*	Mean ± SD or % (*n*)	*n*	Mean ± SD or % (*n*)
Study center				
Poitiers	616	48 (298)	610	47 (284)
Children characteristics				
Age (y)	616	10.7 ± 0.8	610	10.7 ± 0.8
Boys	616	51 (313)	610	51 (313)
Gestational age (wk)	616	39.3 (1.6)	610	39.3 (1.6)
Birthweight (kg)	616	3.3 ± 0.5	610	3.3 ± 0.5
Overweight at the 9–12-y follow-up (IOTF) ^2^	601	10 (63)	596	10 (58)
Maternal characteristics				
Age at delivery (y)	616	30 ± 4.5	610	30.3 ± 4.5
Primiparous	615	48 (298)	609	47 (289)
Pre-pregnancy BMI	607		600	
<18.5 kg/m^2^		8 (48)		8 (45)
18.5–24.9 kg/m^2^		68 (411)		69 (415)
25.0–29.9 kg/m^2^		18 (109)		17 (104)
≥30.0 kg/m^2^		6 (39)		6 (36)
Educational attainment	613		607	
< high school diploma		16 (99)		14 (88)
high school diploma		17 (104)		16 (98)
2-y university degree		25 (153)		25 (153)
>2-y university degree		42 (257)		44 (268)
Monthly household income	612		606	
<€1501		8 (46)		6 (39)
€1501–2300		27 (166)		26 (160)
€2301–3000		31 (190)		31 (186)
>€3000		34 (210)		36 (221)
Infant feeding practices				
Any breastfeeding duration	616		610	
Never		26 (158)		23 (143)
<3.0 months		28 (173)		26 (159)
3.0–5.9 months		29 (179)		27 (167)
≥6.0 months		17 (106)		23 (141)
Age at complementary feeding introduction	616		610	
<4.0 months		27 (167)		25 (152)
4.0–6.0 months		71 (438)		72 (442)
>6.0 months		2 (11)		3 (16)

^1^ Values before multiple imputation. ^2^ Overweight (including obesity) defined using IOTF cut-off [36].

**Table 2 nutrients-13-02659-t002:** Dietary intakes in infancy and the liking scores in young children according to child’s sex ^1^.

	8 Months (*n* = 616)				12 Months (*n* = 610)			
	Total	Boys	Girls	*p*-Value ^2^	Total	Boys	Girls	*p*-Value ^2^
Dietary intake in infancy								
Energy intake (kcal)	710 ± 131	735 ± 138	684 ± 118	<0.0001	798 ± 137	826 ± 136	769 ± 134	<0.0001
Carbohydrate intake (g/day)	98 ± 20	102 ± 20	94 ± 18	<0.0001	108 ± 22	112 ± 22	104 ± 22	<0.0001
Sugar intake (g/day)	58 ± 18	59 ± 18	58 ± 17	0.21	69 ± 16	71 ± 16	67 ± 16	0.0007
Consumer of added sugars	27 (165)	29 (91)	24 (74)	0.19	36 (219)	37 (116)	35 (103)	0.54
Fat intake (g/day)	28 ± 7	28 ± 7	27 ± 7	0.06	30 ± 8	31 ± 8	29 ± 8	0.009
Consumer of added fats	13 (83)	15 (47)	12 (36)	0.25	23 (141)	22 (70)	24 (71)	0.65
Liking scores at 8–12-year-old (0–10)								
Sweetness	4.6 ± 1.3	4.7 ± 1.3	4.4 ± 1.3	0.004	4.6 ± 1.3	4.8 ± 1.3	4.4 ± 1.3	0.004
Fattiness	5.7 ± 1.2	5.9 ± 1.8	5.7 ± 1.7	0.18	5.7 ± 1.2	5.9 ± 1.8	5.7 ± 1.6	0.21
Sweetness-and-fattiness	5.8 ± 1.7	5.8 ± 1.3	5.6 ± 1.2	0.04	5.8 ± 1.7	5.8 ± 1.3	5.7 ± 1.1	0.13

^1^ Values are the mean ± SD or % (*n*). Values before multiple imputation for the dietary intakes. ^2^ Student’s *t*-test and Chi-square.

**Table 3 nutrients-13-02659-t003:** Associations (β (CI 95%)) between infant dietary intakes and their liking for sweetness score in young children.

	Unadjusted	Adjusted ^1^ Imputed	Adjusted ^1^ Complete-Case
Dietary intake at			
8 months ^2^	*n* = 616	*n* = 616	*n* = 555
Carbohydrate intake (per 10 g/d)	0.05 [0.00; 0.11]	0.03 [−0.02; 0.09]	0.02 [−0.04; 0.08]
Sugar intake (per 10 g/d)	0.06 [0.00; 0.12]	0.05 [−0.01; 0.12]	0.05 [−0.01; 0.12]
Consumer of added sugars	−0.01 [−0.26; 0.23]	−0.04 [−0.28; 0.20]	−0.05 [−0.31; 0.21]
12 months ^2^	*n* = 610	*n* = 610	*n* = 543
Carbohydrate intake (per 10 g/d)	Boys: −0.05 [−0.12; 0.01]	Boys: −0.06 [−0.13; 0.01] ^3^ Girls: 0.07 [0.00; 0.14]	Boys: −0.08 [−0.15; −0.01]
	Girls: 0.08 [0.01; 0.15]		Girls: 0.07 [0.00; 0.15]
Sugar intake (per 10 g/d)	Boys: 0.01 [−0.08; 0.11]	Boys: −0.01 [−0.10; 0.09] ^3^ Girls: 0.09 [−0.01; 0.19]	Boys: −0.02 [−0.12; 0.07]
	Girls: 0.11 [0.02; 0.21]		Girls: 0.07 [−0.03; 0.18]
Consumer of added sugars	0.01 [−0.21; 0.23]	0.02 [−0.20; 0.24]	−0.04 [−0.28; 0.20]

^1^ Linear regression on multiple imputed datasets (*n* = 5) also adjusted for study center, sex, age, gestational age, birthweight, maternal characteristics (age at delivery, pre-pregnancy BMI, educational attainment and maternal dietary patterns during pregnancy), household income, parity, any breastfeeding duration and age at complementary food introduction. ^2^ One separate model was run for each dietary exposure. ^3^
*P*-values for interaction with sex tests were 0.003 and 0.07 for associations between carbohydrate intake at 12 months, sugar intake at 12 months and liking for sweetness, respectively. All other models, *p*_interaction_ > 0.10.

**Table 4 nutrients-13-02659-t004:** Associations (β (CI 95%)) between infant dietary intakes and their liking for fattiness scores in young children.

	Unadjusted	Adjusted ^1^ Imputed	Adjusted ^1^ Complete-Case
Dietary intake at			
8 months ^2^	*n* = 616	*n* = 616	*n* = 555
Fat intake (per 10 g/d)	0.07 [−0.07; 0.21]	0.02 [−0.12; 0.16]	0.03 [−0.12; 0.17]
Consumer of added fats	−0.01 [−0.3; 0.28]	−0.03 [−0.32; 0.27]	0.08 [−0.24; 0.41]
12 months	*n* = 610	*n* = 610	*n* = 543
Fat intake (per 10 g/d)	0.07 [−0.05; 0.18]	0.04 [−0.08; 0.16]	0.05 [−0.08; 0.17]
Consumer of added fats	−0.09 [−0.32; 0.14]	−0.05 [−0.28; 0.18]	−0.12 [−0.38; 0.13]

^1^ Linear regression on multiple imputed datasets (*n* = 5) also adjusted for study center, sex, age, gestational age, birthweight, maternal characteristics (age at delivery, pre-pregnancy BMI, educational attainment and maternal dietary patterns during pregnancy), household income, parity, any breastfeeding duration and age at complementary food introduction. ^2^ One separate model was run for each dietary exposure.

**Table 5 nutrients-13-02659-t005:** Associations (β (CI 95%)) between infant dietary intakes and their liking for fattiness-and-sweetness scores in young children.

	Unadjusted	Adjusted ^1^ Imputed	Adjusted ^1^ Complete-Case
Dietary intake at			
8 months ^2^	*n* = 616	*n* = 616	*n* = 555
Carbohydrate intake (per 10 g/d)	0.07 [0.00; 0.14]	0.09 [0.01; 0.16]	0.07 [−0.01; 0.14]
Sugar intake (per 10 g/d)	0.04 [−0.04; 0.12]	0.04 [−0.04; 0.12]	0.04 [−0.05; 0.12]
Consumer of added sugars	−0.06 [−0.46; 0.35]	−0.16 [−0.48; 0.15]	−0.18 [−0.51; 0.16]
Fat intake (per 10 g/d)	0.01 [−0.18; 0.20]	−0.03 [−0.23; 0.17]	−0.01 [−0.21; 0.20]
Consumer of added fats	−0.13 [−0.44; 0.18]	−0.05 [−0.47; 0.36]	0.06 [−0.40; 0.32]
12 months	*n* = 610	*n* = 610	*n* = 543
Carbohydrate intake (per 10 g/d)	0.03 [−0.03; 0.09]	0.02 [−0.04; 0.08]	0.01 [−0.06; 0.07]
Sugar intake (per 10 g/d)	0.01 [−0.07; 0.10]	0.00 [−0.09; 0.09]	−0.01 [−0.10; 0.08]
Consumer of added sugars	−0.02 [−0.34; 0.31]	0.04 [−0.25; 0.32]	0.03 [−0.28; 0.33]
Fat intake (per 10 g/d)	0.00 [−0.16; 0.17]	−0.02 [−0.18; 0.15]	−0.02 [−0.19; 0.15]
Consumer of added fats	0.00 [−0.28; 0.29]	0.03 [−0.30; 0.36]	−0.04 [−0.40; 0.32]

^1^ Linear regression on multiple imputed datasets (*n* = 5) also adjusted for study center, sex, age, gestational age, birthweight, maternal characteristics (age at delivery, pre-pregnancy BMI, educational attainment and maternal dietary patterns during pregnancy), household income, parity, any breastfeeding duration and age at complementary food introduction. ^2^ One separate model was run for each dietary exposure.

## Data Availability

The data underlying the findings cannot be made freely available for ethical and legal restrictions imposed, because this study includes a substantial number of variables that, together, could be used to re-identify the participants based on a few key characteristics and then be used to have access to other personal data. Therefore, the French ethics authority strictly forbids making these data freely available. However, they can be obtained upon request from the EDEN principal investigator. Readers may contact barbara.heude@inserm.fr to request the data. The analytic code will be made available upon request pending application and approval.

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
