# Peer review of "Associations between Infant Dietary Intakes and Liking for Sweetness and Fattiness Sensations in 8-to-12-Year-Old Children"

_nutrients, 2021, doi:10.3390/nu13082659_

Round 1
Reviewer 1 Report
Dear Authors,
The data presented herein are underwhelming, and it is not clear how data could be collected from food records, which may have recall bias. This data should be collected prospectively.
Based on introduction, (lines 46-47), why did you consider that you would see any differences? The rationale for this work is not clear.
Why would there be a sex-based differences?
Sample size is likely sufficient, however unless a sample size calculation was done, and depending on the effect size, perhaps this sample size needed to be even larger?
The results presented are not significant enough for publication in Nutrients.
Author Response
Dear Authors,
The data presented herein are underwhelming, and it is not clear how data could be collected from food records, which may have recall bias. This data should be collected prospectively.
Response:
We are sorry for not having reported this more clearly. First, dietary intakes were collected during infancy. Actually, the instructions given to parents to complete the food records mentioned that the food record should be completed in a daily basis as we asked the exact time of each food intake at 8 and 12 months of age. Second, when aged between 8 and 12 years of age, children filled the online food liking questionnaire. Hence, liking for sweetness, sweetness-and-fattiness, and fattiness were not measured concomitantly with infant dietary intakes. We apologies if our methods section was not explicit enough. We have switched the section 2.2 and 2.3 and added clarifications in our Methods section to help the understanding:
Lines 104-106 : “Between May and September 2015 (when the children were aged 8 to 12 years), par-ents were invited to participate either by electronic mail (78 %) or by paper mail (22 %), when no valid electronic mail address was available.”
Lines 108-110 : “Infant food intake was reported on a daily basis (according to the instructions) by mothers using a 3-day food record including two weekdays and one weekend day when their child was aged 8 and 12 months”
Lines 124-125: “Children’s liking for sweetness and fattiness sensations was assessed when they were aged between 8 and 12 years by a specific and validated questionnaire”.
Based on introduction, (lines 46-47), why did you consider that you would see any differences? The rationale for this work is not clear.
Response:
As Okronipa et al (1, 2) mentioned previously, the level of sugar exposure in their randomized trial might be too low to observe an association with later sweetness preference or sugar consumption. As we cannot extrapolate these results to different level of sugar and as previous evidence (3) suggested an association between higher exposure to sugar in infancy and higher preference for higher level of sucrose (in liquid solution), we were interested to study whether this positive association can be observed in other study populations. We believe that replicating studies in different setting/study population is necessary. We explained the rationale of our study in the introduction section Lines 45-72:
“In contrast to liking for sweetness which is innate [27], liking for fattiness is not congenital but rather learned through the association between the ingestion of fatty foods and its positive physiological consequences (satiety) [28-33]. However, whether high levels of sugar intake in infancy could contribute to higher liking for sweetness in later childhood is ambiguous [34]. Previous findings suggested that repeated exposures to sweetened foods can be part of the learning process of liking for sweetness [16, 35, 36]. A longitudinal study conducted in the US, has shown that repeated exposures to sweetened beverages during infancy could lead to a preference for solutions with higher concentrations of sucrose later in childhood [16]. In a randomized trial, expo-sure to a slightly sweet supplement from 6 to 18 months was neither associated with later preference, consumption of sweet foods and beverages [15] nor with the level of sweet taste most preferred in 4- to 6-y-old Ghanaian children [14]. The authors hypothesized that the sugar content in their supplement could have been too low to shift the children’s behavior towards sweetened foods at later stages [14, 15] compared with the previous study on frequent consumption of sweetened beverages [16], because of the highly and routine degree of exposure to sweetened foods, in particular sweetened beverages during the first two years of life [14, 15, 37]. However, this trial did not investigate the influence of overall dietary exposure to sugar during infancy on later liking for sweetness in mid-childhood but only the influence through the supplementation. Similarly, the American study has only investigated the influence of consumption of sweetened beverages in infancy, but not the total sugar intake [16]. Hence, there is no evidence on whether infant overall sugar intake could influence later liking for sweetness. Furthermore, except for the two aforementioned longitudinal studies, there is a lack of evidence on the long-term association between early dietary exposure to sugar and mid-childhood sweetness preference. Therefore, there is a need to investigate further this association in prospective studies and in population settings with different level of sugar exposure. Concerning fattiness, this sensation is contributing to food palatability which can contribute to overconsumption [38], but the influence of early exposure on liking for fattiness sensation was not documented. “
Why would there be a sex-based differences?
Response:
Sexual dimorphism is a well-known phenomenon identified in research and large amount of evidence from multiple domains (genetic, physiology, psychology/behavior...) has supported this (4-6). Keller et al. have brought out that there is still a need for further investigations of the expression of sexual dimorphism in the development of eating behaviors as it remains poorly explored. Nevertheless, some evidence drove our hypothesis on the differential development of eating behaviors regarding early sex-based differences in dietary intakes. As mentioned in our Introduction section, boys showed greater activation to visual food cues in the right hippocampus and visual cortex than girls (7). However, girls might be more sensitive to sweet taste (8, 9). In other terms, similar dietary exposure among boys and girls could be perceived differently and lead to different reactions depending on child sex. We have stated this Lines 73-84 in the Introduction section:
“Sex-related differences in dietary behavior have already been noted in children, regarding both food intake [39, 40] and food preferences [39, 41-43]. Previous evidence suggested that infant feeding practices differed by sex. In France for instance, girls were more likely to be introduced later to complementary food, breastfed longer and consumed more ready prepared baby foods than boys [44-46] which may lead to a lower exposure to sweetness and fattiness in infancy [40]. Furthermore, girls tended to consume more and have greater liking for fruit and vegetables than boys [39, 41-43]. In contrast, boys had higher liking for sweetened and fatty foods than girls [41]. Neural responses related to food cues were also evidenced in children. Boys showed greater activation to visual food cues in the right hippocampus and visual cortex than girls [47]. In contrary, girls might be more sensitive to sweet taste [48, 49]. In other words, the same dietary exposure among boys and girls could be perceived differently and lead to different behavioral reactions depending on sex.”
Sample size is likely sufficient, however unless a sample size calculation was done, and depending on the effect size, perhaps this sample size needed to be even larger?
Response:
This study was conducted as an ancillary study in a cohort; for this reason, we did not perform an a priori sample size calculation. The effect size, if any, is of small amplitude.
We believe that even if we had a larger sample size within the same country, the conclusion might remain the same, in relation to the low level of dietary exposure in infancy regarding sugar (27 % and 36% of consumer of added sugars at 8 and 12 months, respectively) and fat intake (13% and 23 % of consumer of added fat at 8 and 12 months, respectively) in the studied population (highly educated French mothers). This low level of exposure and somewhat lack of heterogeneity is indicated in Table 2 and further discussed in the Discussion section. Our study needs to be replicated in population where level of exposure to dietary sugar and fats is more varied, and would reach higher levels (3, 10, 11). We added the following sentences Lines 330-335 in the Discussion section;
“The lack of association in our study could be partly due to the relatively low and homogeneous level of exposure in infancy in our study sample (about 60 and 70g per day of sugar intake at 8 and 12 months, respectively and about 30g per day of fat intake at both 8 and 12 months). Further studies have to be conducted in population with larger variation in exposure level and probably higher consumption of added sugars and fats “
The results presented are not significant enough for publication in Nutrients.
Response:
We acknowledge that our analysis did not highlight strong associations. However, current nutritional guidelines recommend to limit sugar intake in infancy but to provide sufficient lipids during complementary feeding (12, 13). Our analyses indicated that low to moderate exposure to fat in infancy does not seem to have an impact on liking for fattiness sensation in mid-childhood. For sugar exposure, even if no strong association was found, our results could reinforce current recommendation of limiting sugar intake in early life, in order to limit sweetness liking at later stage in life. Further studies are needed to examine the influence of high exposure to sugar and fat in infancy on liking to sweetness or fattiness sensations.

Reviewer 2 Report
Thank you for the opportunity to review this manuscript.
I have one small suggestion to improve readability. Could you please move the information about child age (line 204-205) to both the abstract and section 2.1?
For example line 15 "The liking score (0-10) for the different sensations was assessed through an online child-completed questionnaire at mean age 10.7 years", and add "At this time children were aged from 8.7 to 12.5 years" around line 76.
At first reading of 5-y follow up on line 76, I was a bit confused. I was wondering how children could complete an online questionnaire if they were only five years old!
Author Response
Thank you for the opportunity to review this manuscript.
I have one small suggestion to improve readability. Could you please move the information about child age (line 204-205) to both the abstract and section 2.1?
For example line 15 "The liking score (0-10) for the different sensations was assessed through an online child-completed questionnaire at mean age 10.7 years", and add "At this time children were aged from 8.7 to 12.5 years" around line 76.
At first reading of 5-y follow up on line 76, I was a bit confused. I was wondering how children could complete an online questionnaire if they were only five years old!
Response:
We thank the reviewer for this suggestion and apologies for the misleading sentence. We added the child age information in both the abstract and section 2.1 (Lines 14 and 104-105).

Reviewer 3 Report
Thank you for such a fine study! There are a number of clarifications that are recommended:
Introduction: line 36: Please add a new paragraph that describe the definitions and the scientific difference, with appropriate scientific references, between 'preference' and 'liking' in general and then use sweetness as an example.
Methods: line 175 ff under statistical analysis: The first sentence in this section is clear (lines 173-175) however the subsequent section through line 187 is not. Please reword this entire section for clarity. While it may seem repetitive to you as authors, as readers it is hard to understand the actual steps that were taken and 'exposure variables' and what were the "a priori hypotheses". Please restate these as clarity is more important than whether there is repetitiveness.
Discussion: line 287: "However, other studies did not find such associations." please add several references at the end of this sentence.
line 309" " older children in the US. Please add the word the here.
Author Response
Thank you for such a fine study! There are a number of clarifications that are recommended:
Introduction: line 36: Please add a new paragraph that describe the definitions and the scientific difference, with appropriate scientific references, between 'preference' and 'liking' in general and then use sweetness as an example.
Response:
We thank the reviewer for this suggestion. We added the following paragraph in the Introduction section lines 39-44:
“Taste preference is usually referring to the comparison of individual reaction to different concentrations of tastants (sucrose, oleic fatty acid…) or to a tastant relative to water [14-19], whereas taste liking is rather referring to the degree of appreciation towards a food taste on a hedonic scale [20-25]. As an example, two individuals can both prefer high level of sweetness to low level of sweetness; but one can rate degree of liking for high level of sweetness higher than the other [26].
Methods: line 175 ff under statistical analysis: The first sentence in this section is clear (lines 173-175) however the subsequent section through line 187 is not. Please reword this entire section for clarity. While it may seem repetitive to you as authors, as readers it is hard to understand the actual steps that were taken and 'exposure variables' and what were the "a priori hypotheses". Please restate these as clarity is more important than whether there is repetitiveness.
Response:
We thank the reviewer for raising this issue. We have changed the previous paragraph as follows Lines 202-218:
“Unadjusted and adjusted associations between dietary exposure variables in infancy and liking outcome variables in mid-childhood were tested either using linear or logistic regressions. Each dietary intake variable (carbohydrate, sugar or fat intake, being a consumer of added sugars or added fats) was tested individually, i.e. not concomitantly with another dietary variable within the same model. Carbohydrate intake, sugar intake and consumption of added sugars were consecutively analyzed in association with liking for sweetness sensation. Fat intake and consumption of added fats were successively investigated in relation with liking for fattiness sensation. Finally, carbohydrate, sugar or fat intake, being a consumer of added sugars or added fats were consecutively studied in association with liking for fattiness-and-sweetness sensation. For all the studied associations, potential confounding factors were identified in the literature and selected using the directed acyclic graph (DAG) method [39]. Hence, the following set of confounding variables were considered for all models: study centre, children’s characteristics (sex, age, birthweight, gestational age), maternal characteris-tics (age, pre-pregnancy BMI, educational attainment, household income, parity and maternal scores on healthy and western dietary patterns during pregnancy), any breastfeeding duration and age at complementary food introduction”
Discussion: line 287: "However, other studies did not find such associations." please add several references at the end of this sentence.
Response:
We thank the reviewer for this suggestion. We added the relevant references at the end of this sentence.
line 309" " older children in the US. Please add the word the here.
Response:
We thank the reviewer for noticing this oversight, we have corrected it in the paper.
